# Phthalylglycyl Chloride as a Derivatization Agent for UHPLC-MS/MS Determination of Adrenaline, Dopamine and Octopamine in Urine

**DOI:** 10.3390/molecules28072900

**Published:** 2023-03-23

**Authors:** Maria Zorina, Victor V. Dotsenko, Pavel N. Nesterenko, Azamat Temerdashev, Ekaterina Dmitrieva, Yu-Qi Feng, Sanka N. Atapattu

**Affiliations:** 1Analytical Chemistry Department, Kuban State University, 149 Stavropolskaya St., 350040 Krasnodar, Russia; 2Department of Organic and Analytical Chemistry, North-Caucasus Federal University, 1 Pushkina St., 355000 Stavropol, Russia; 3M.V. Lomonosov Moscow State University, Leninskie Gory 1, 119991 Moscow, Russia; 4Department of Chemistry, Wuhan University, Wuhan 430072, China; 5CanAm Bioresearch Inc., Winnipeg, MB R3T 6C6, Canada

**Keywords:** phthalylglycyl chloride, catecholamines, dopamine, adrenaline, octopamine, derivatization

## Abstract

Dopamine, adrenaline and octopamine are small polar molecules that play a vital role in regulatory systems. In this paper, phthalylglycyl chloride was proposed as a derivatization agent for octopamine, adrenaline and dopamine determination in urine for the first time. The derivatization procedure facilitated the use of reversed-phase liquid chromatography with positive electrospray ionization–high-resolution mass spectrometry. An LC-HRMS method was developed that provided quantification limits of 5 ng/mL and detection limits of 1.5 ng/mL for all analytes. The 95–97% yield of derivates was observed after a 10 min derivatization with phthalylglycyl chloride at pH 6.5 and 30 °C. The proposed method was successfully applied to the analysis of human urine samples. The obtained results were compared with those of conventional derivatization procedures with 9-fluorenyl-methoxycarbonyl chloride and dansyl chloride.

## 1. Introduction

Dopamine, adrenaline and octopamine are important catecholamines that play a vital role in regulatory functions and can be used in Alzheimer’s disease diagnostics. They are also involved in mediating a number of cognitive functions, spatial, recognition and working memory [1,2,3]. According to published research [1,2,3,4], their concentration below 5 ng/mL could be associated with functional changes and neuropathology, which makes their accurate and precise determination an actual task.

The studied compounds contain phenol- or catechol- groups and a side primary or secondary amino- group (Figure 1). Due to their high hydrophilicity and basicity (see the properties in Table 1), these compounds can be effectively separated by various techniques including ion chromatography (IC) [5], hydrophilic chromatography (HILIC) [6] or ion-pair reversed-phase HPLC using the addition of anionic surfactants to the eluent [7,8]. It should be noted that native catecholamines are not retained on common chromatographic columns for reversed-phase HPLC, so derivatization can be used to increase their hydrophobic properties.

The most used methods for their determination are liquid chromatography with tandem mass spectrometry (LC-MS/MS) [1,2,3,4], liquid chromatography with electrochemical detection (LC-ECD) [13,14,15] and liquid chromatography with ultraviolet (LC-UV) or fluorometric detection (LC-FLD) [16,17,18,19,20].

However, the recent advances in the synthesis of highly efficient sorbents make possible the usage of so-called “polar” C_18_ columns with a sub-3 µm particle size, which provides greater retention for small polar molecules [21]. At the same time, mixed-mode C_18_ columns with the same particle size available on the market nowadays could be a suitable solution [22]. They also provide a significant increase in the retention of some polar compounds because of the combination of the conventional reversed-phase and weak cation-exchange separation mechanisms [23]. They are also suitable for the separation of some critical pairs for conventional C_18_ sorbents. The main limitation for such columns is the prevalence of the conventional columns for multicomponent screening in most routine laboratories, and application of the novel sorbents requires at least method revalidation.

In the study [22], possible ways of using a mixed-mode sulfonate-modified sorbent for catecholamines’ separation were demonstrated. According to their conclusions, due to the coexistence of reversed-phase and strong coulombic interactions, polar cationic compounds have significantly higher retention and much better separation through a “hydrophobically assisted cation-exchange mechanism”. The most interesting part of the research is the comparison of the conventional C_18_ and silica-based hyper-crosslinked sulfonate-modified reversed phase and the clear demonstration of its advantages. Significant differences in the selectivity and efficiency of other compounds’ separation were also noticed based on the results of 47 drugs’ separation.

Another important methodological aspect is that a lot of different combinations for mobile phase compositions and ion-pair reagents are applicable for UV or FLD detection, which could be used to increase analytes’ retention in the RP-HPLC separation mode with conventional columns. In the case of MS detection, the number of possible mobile phases and their modifications is extremely limited [24].

In the case of MS detection, a possibility of strong matrix effects and high chemical noise level could be observed and should be evaluated. Prior to analysis, derivatization is usually carried out for several reasons: firstly, it allows their polarity to be decreased making possible separation by reversed-phase liquid chromatography; secondly, it results in an increase in their molecular weight, which reduces the signal-to-noise ratio in mass spectrometric detection; and thirdly, it can make fluorescent detection possible due to the addition of the respective functional groups. Typically, 9-fluorenyl-methoxycarbonyl chloride (FMOC-Cl), (DNS-Cl) [1] and benzoyl-chloride [25] are used for these purposes.

Overall, derivatization [2,3] is preferable for the determination of such polar and small molecules as catecholamines. The above-mentioned reagents, namely, FMOC-Cl and dansyl chloride, require at least 30 min for reaction and the presence of an alkali [18,19]. It leads to a high degradation speed of some analytes (including catecholamines) and requires neutralization of the alkali in a final step of the sample preparation procedure.

The most used reagents for derivatization and methods for catecholamines’ determination are shown in Table 2.

Due to their limitations, the synthesis of novel inexpensive derivatization agents that could provide sufficient sensitivity, selectivity, chromatographic peak shape, quick derivatization at close to neutral pH value and specific MS fragmentation has become an actual task.

Phthalylglycyl chloride (PG-Cl) is a commonly used reagent for amido-alkylation for the synthesis of benzofuran derivatives, remote C-H bond functionalization and photoinduced single electron transfer cyclization for the preparation of cyclic peptides [33,34,35,36]. At the same time, it could be used as a derivatization reagent for amines. It features a phthalimide fragment, which improves retention on reversed-phase columns, and the presence of additional nitrogen in the structure could positively affect ionization efficiency of the analytes. Moreover, it should be noticed that the PG-fragment in derivatives is not as hydrophobic and small in comparison with FMOC-Cl and DNS-Cl, and even minor differences in the native analytes make it easier to separate them.

Low stability of these reagents in aqueous solutions limits their application. To prevent it, sample dilution with acetonitrile is required. It could also be used for protein precipitation as a step in the sample preparation procedure [37].

The aim of this research is the investigation of the selectivity and efficiency of PG-Cl as a derivatization agent for catecholamines’ determination in urine.

## 2. Results

### 2.1. Phthalylglycyl Chloride (PG-Cl) Synthesis

*N*-phthaloylglycine was prepared in 95% yield by the known procedure [38], based on the reaction of glycine with phthalic anhydride in a hot DMF solution for 6 h (Figure 2). The spectral data are in accordance with those reported in [39].

*N-phthaloylglycine chloride* (**1**) was prepared by a slightly modified procedure [33] as follows: *N*-phthaloylglycine (2.052 g, 0.01 mol) was treated with an excess of SOCl_2_ (5.0 mL) in a moisture-free atmosphere. The resulting mixture was stirred until the evolution of gases ceased (about 3 h) and then allowed to stand overnight. The excessive thionyl chloride was then evaporated in a vacuum, and the solid residue was treated with hexane. Hexane was re-evaporated, and this process was repeated once more to remove trace amounts of SOCl_2_. The resulting white crystalline solid was filtered off and washed on a filter with petroleum ether, and dried in a vacuum to afford pure *N*-phthaloylglycine chloride PG-Cl. The yield was 89%. ^1^H NMR (400 MHz, CDCl_3_), δ 4.85 (s, 2H, CH_2_), 7.77–7.80 (m, 2H, H5, H6), 7.90–7.94 (m, 2H, H4, H7). ^13^C NMR (101 MHz, CDCl_3_), δ 44.5 (CH_2_), 124.0 (2 CH), 131.5 (2 CH), 134.8 (2C), 167.0 (2C = O), 169.0 (C = O).

### 2.2. Optimization of Derivatization Conditions

In previous studies [37,39], different approaches utilizing FMOC-Cl and DNS-Cl were discussed, and optimal conditions for derivatization were established. However, PG-Cl was not previously described as a derivatization reagent, and its selectivity and reaction speed as well as the analytical characteristics of its derivatives are unknown.

The main aim of acetonitrile usage as a solvent for PG-Cl was its high hydrolysis speed. Dry acetonitrile is a commonly used and available solvent suitable for further reversed-phase separation. In our study, parameters such as pH value (established optimum value was 6.5), temperature (Figure 3) and reaction time (Figure 4) were optimized for dopamine and octopamine in synthetic urine, prepared according to [40]. As can be seen from the presented figures, the maximum yield of derivatization was observed at 30 °C after 10 min of the reaction. A further increase in the temperature leads to degradation of the derivatives, which could not be inhibited by changing the pH value or the addition of such reagents as methylamine.

The stability of the analytes at 30 °C is an important part of the investigation because it shows a possibility for further adaptation for routine analysis. When the reaction was finished, samples were transferred into the glass vials and stored at 5 °C in an autosampler tray prior to analysis, and they were found to be stable up to 24 h after their preparation (RSD < 15%).

During the method optimization, the high speed of the reagent hydrolysis in an aqueous solution was established, and its low stability requires the usage of a freshly prepared solution before the experiments. An important factor determining the stability of the obtained derivatives was the pH value. The derivatization reaction was inhibited in an acidic solution, leading to the prevalence of the hydrolysis process of the derivatization agent, while in an alkaline solution, the hydrolysis speed predominated over the formation of derivatives. The derivatives were stable when the reaction was carried out at pH 6.5.

### 2.3. MS Detection

The investigation of PG-Cl suitability for LC-MS/MS and LC-HRMS was conducted with two major points: informative fragments produced in the collision cell of the mass spectrometer and the influence of the derivative fragment on retention parameters.

Typically, one of the most abundant product ions observed in the MS/MS spectrum is a part of the derivatization agent (Figure 5). In this case, the same result was observed. A phthaloylglycyl moiety of the molecule was easily eliminated at low collision energies (up to 7 eV for the investigated compounds) producing the abundant fragment ion with *m*/*z* 160.0393. All the other fragments corresponded to typically observed fragments for native catecholamines.

### 2.4. Chromatographic Conditions

As for chromatographic properties, the proposed derivatization agent significantly increased the retention of the catecholamines on a reversed-phase column (Figure 6) and allowed the use of a conventional screening gradient elution program with a high speed of mobile phase composition change.

A significant difference in the retention times between analytes and the hydrolyzed form of the derivatization reagent makes it possible to use data-dependent and data-independent acquisition modes with low collision energies without warnings about overlapping of the product ions in non-targeted screening applications. As can be seen from Figure 6, octopamine and dopamine peaks overlapped by less than 10% under the described gradient elution conditions, which does not lead to a higher standard deviation in quantitative analysis.

Another important aspect of the proposed method is establishing some metrological characteristics, which could be obtained with PG-Cl for catecholamines’ determination (Table 3).

Calibration solutions were used to establish linear ranges, while quality control solutions were analyzed to evaluate inter- and intra-day accuracy and precision. Accuracy was assessed as the bias between the nominal and observed concentrations within one day and on different days. Precision was expressed as the relative standard deviation of the results.

## 3. Discussion

### 3.1. Comparison of PG-Cl with FMOC-Cl and DNS-Cl

In comparison to conventional derivatizing agents such as FMOC-Cl or DNS-Cl, PG-Cl is not a fluorescent marker. However, the presence of a secondary nitrogen atom promotes its use as a derivatization agent for LC-MS since it provides a higher ionization yield in comparison to FMOC-Cl and DNS-Cl and improves the detection and quantification limits obtained using synthetic urine samples (Table 4).

As can be seen from Table 4, PG-Cl provides up to a two times higher quantification limit for all analytes and demonstrates a notable difference in the detection limit for dopamine, which can be due to a significant difference in the ionization efficiency of the applied derivatization reagents and, as a result, in the derivatives.

### 3.2. Urine Samples Analysis

Real urine samples were prepared according to Section 2.4; the results obtained using FMOC-Cl and PG-Cl as derivatization reagents are provided in Table 5. The good convergence of the results suggests that the proposed approach may be promising for further research. This section may be divided by subheadings. It should provide a concise and precise description of the experimental results and their interpretation, as well as the experimental conclusions that can be drawn.

## 4. Materials and Methods

### 4.1. Chemicals

Standards of dopamine (≥98%), octopamine (≥98%), adrenaline (epinephrine) (≥99%), 9-fluorenyl-methoxycarbonyl chloride (≥99%), dansyl chloride (≥99%) were purchased from Sigma-Aldrich (St. Louis, MO, USA), gabapentin (IS) (≥75%) was obtained from Pfizer (New York, NY, USA). HPLC-grade acetonitrile (“Biosolve”, Jerusalem, Israel), 18.2 MΩ water (Milli-Q, Millipore, Molsheim, France) and formic acid (98%, Acros Organics, Geel, Belgium) were used as the mobile phase. Methanol of HPLC grade was purchased from Vecton (Saint-Petersburg, Russia). Potassium carbonate (≥99%, Vecton, Saint-Petersburg, Russia), potassium bicarbonate (≥99%, Vecton, Saint-Petersburg, Russia), sodium hydroxide (≥99%, Reactive, Saint-Petersburg, Russia), sodium tetraborate (≥99%, Vecton, Saint-Petersburg, Russia), sodium hydrogen phosphate (≥99%, Vecton, Saint-Petersburg, Russia), potassium dihydrogen phosphate (≥99%, Vecton, Saint-Petersburg, Russia), ammonium acetate (≥99%, Vecton, Saint-Petersburg, Russia) and acetic acid were used for the preparation of buffer solutions with pH 10.5, 9.5, 6.5 and 4.5, respectively. Glycine (HPLC grade, Vecton, Saint-Petersburg, Russia), phthalic anhydride (99%, Vecton, Saint-Petersburg, Russia), dimethylformamide (HPLC grade, Vecton, Saint-Petersburg, Russia) were used for PG-Cl synthesis.

### 4.2. Instrumentation

A Bruker MaXis Impact (Bruker Daltonik GmbH, Bremen, Germany) quadrupole-time-of-flight mass spectrometer (Q-TOF) equipped with an electrospray ionization (ESI) source coupled with an ultra-high performance liquid chromatography Bruker Elute system (UHPLC) with a Phenomenex Kinetex C18 (100 mm × 2.1 mm, 1.7 μm) column and an appropriate guard column was used for the chromatographic separation. A two-component system of methanol (A)–0.1% formic acid in water was (B) used as the mobile phase. The gradient elution program was as follows: 0.0–1.0 min 5% A; 2.7–4.0 min 60% A; 5.0–7.5 min 90% A; 7.51–9.0 min 5% A. The injection volume was 10 μL. The flow rate was held constant at 0.45 mL/min, and the column thermostat temperature was 40 °C. The voltage at the ionization source was 3.5 kV, drying gas flow rate was 8 L/min, spray gas pressure was 2 bar, temperature of the ionization source was 250 °C, mass scanning range (*m*/*z*) was 50–600, scanning speed was 3 Hz. Data acquisition and analysis were performed with Bruker Compass HyStar 4.1 and Bruker DataAnalysis 4.4 software, respectively.

### 4.3. Urine Samples

Urine samples obtained from volunteers (males and females aged between 20 and 45) were used to prepare calibration curves and validate the procedure. The samples were preserved with sodium azide and then stored at −20 °C prior to analysis.

### 4.4. Urine Sample Preparation

Due to high concentrations of the analytes in urine samples, 50 µL of urine was transferred into a 1.5 mL Eppendorf tube followed by the addition of 850 µL of water–acetonitrile (30:70, *v*/*v*) mixture containing gabapentin as the internal standard and 100 µL of 250 μg/mL PG-Cl solution in acetonitrile for analytes’ derivatization. After vortex mixing for 2 min, the samples were incubated at 30 °C for 10 min, diluted by 500 µL of water–methanol mixture (1:1, *v*/*v*), centrifuged at 10,000 rpm for 10 min, and 500 µL of the supernatant was transferred into the glass vial. The final concentration of IS in samples was 50 ng/mL.

### 4.5. Preparation of Standard and Stock Solutions

Stock standard solutions of catecholamines (adrenaline, dopamine, octopamine) with the concentrations of 1 mg/mL were prepared by dissolving accurately weighted reagents in acetate, phosphate, borate and carbonate buffers and were further diluted with acetonitrile to obtain working solutions. Working solutions of the derivatization agents (FMOC-Cl, PG-Cl, DNS-Cl at 1 mg/mL) were obtained by dissolving appropriate reagent weights in acetonitrile. Quality control (QC) solutions containing catecholamines at high (250 ng/mL), medium (100 ng/mL) and low (10 ng/mL) concentrations were prepared independently from the working solutions.

## 5. Conclusions

According to the obtained results, PG-Cl can provide greater sensitivity in comparison to FMOC-Cl and DNS-Cl. Considering the high concentrations of catecholamines in real samples, it can be assumed that the sensitivity will be excessive, which makes it possible for a significant sample dilution to increase the robustness and decrease the signal suppression caused by matrix compounds. At the same time, matrices such as plasma and serum contain large amounts of amines, peptides and proteins, which may cause competitive reactions and lead to unstable results of the quantification.

## Figures and Tables

**Figure 1 molecules-28-02900-f001:**
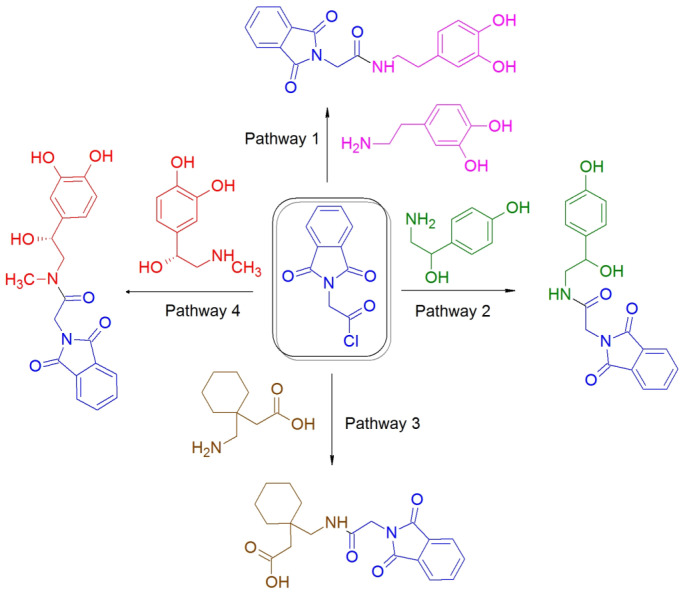
Scheme of catecholamines derivatization with phthalylglycyl chloride (pathway 1—dopamine, pathway 2—octopamine, pathway 4—adrenaline) and pathway 3—gabapentin.

**Figure 2 molecules-28-02900-f002:**
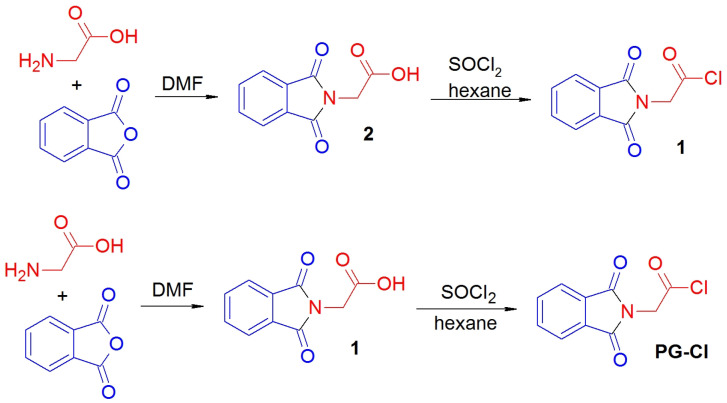
Scheme of *N*-phthaloylglycine chloride synthesis.

**Figure 3 molecules-28-02900-f003:**
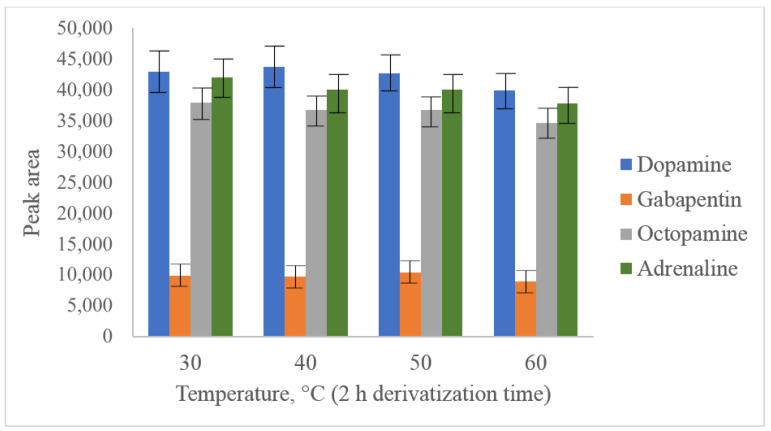
Optimization of derivatization temperature with PG-Cl at 100 ng/mL dopamine, octopamine, adrenaline and 50 ng/mL gabapentin.

**Figure 4 molecules-28-02900-f004:**
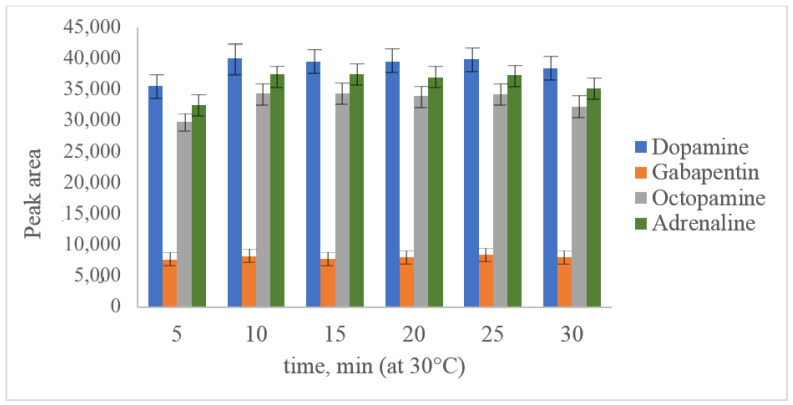
Optimization of derivatization reaction time with PG-Cl at 100 ng/mL dopamine, octopamine, adrenaline and 50 ng/mL gabapentin.

**Figure 5 molecules-28-02900-f005:**
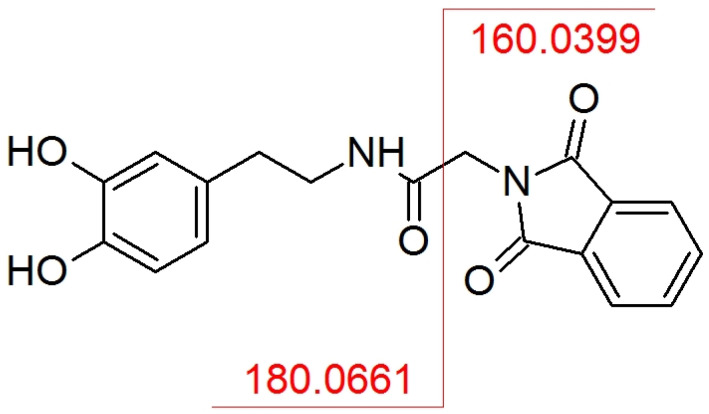
Possible fragmentation method of the most abundant product ion (*m*/*z* 160.0393) for investigated analytes on dopamine example.

**Figure 6 molecules-28-02900-f006:**
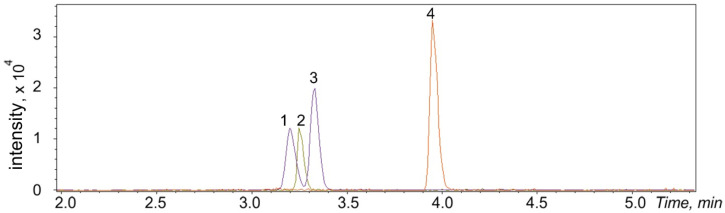
LC-HRMS chromatogram of catecholamines and gabapentin derivatives in synthetic urine at quality control (QC) low (10 ng/mL) concentration: 1—PG-octopamine, 2—PG-adrenaline, 3—PG-dopamine, 4—PG-gabapentin (IS).

**Table 1 molecules-28-02900-t001:** Hydrophobic and acid-base properties of studied biogenic amines.

Name	logP_exp_ (X1)	pK_a1_	pK_a2_	pK_a3_	Reference
Adrenaline	−1.20	8.63	9.87	13.15	[9]
Octopamine	−0.90	8.88	9.53	-	[10]
Gabapentin	−1.10	3.68	10.70		[11]
Dopamine	−0.98	8.81	10.5		[12]

**Table 2 molecules-28-02900-t002:** An overview of some methods for the determination of catecholamines.

Matrix	Analyte	Sample Preparation, Separation Technique	LOQ	Method	Ref.
Plasma	Noradrenaline (NA), adrenaline (A), dopamine (D)	SPE (Strata X-CW), HILIC	NA: 7.4 ng/mL	LC-MS/MS (ESI(+))	[26]
			A: 3.8 ng/mL		
			D: 5.4 ng/mL		
Urine	Normetanephrine (NMN), metanephrine (MN), NA, A, D	SPE (Strata X-CW), HILIC	NMN: 5.0 ng/mL	LC-MS/MS (ESI(+))	[27]
			MN: 5.0 ng/mL		
			NA: 5.0 ng/mL		
			A: 5.0 ng/mL		
			D: 5.0 ng/mL		
Urine	NA, A, D	SPE (Oasis HLB), HILIC	NA: 0.4 ng/mL	LC-MS/MS (ESI(+))	[28]
			A: 0.2 ng/mL		
			D: 0.3 ng/mL		
Urine	NA, A, D	Template imprinted polymers, RP	NA: 157 ng/mL	LC-UV	[29]
			A: 51 ng/mL		
			D: 141 ng/mL		
Urine	NA, A, D	Magnetic iron oxide nanoparticles, RP	NA: 0.21 ng/mL	LC-FLD	[30]
			A: 0.32 ng/mL		
			D: 0.51 ng/mL		
Urine	NA, A, D	Derivatization with phenyl isothiocyanate, RP	NA: 0.10 ng/mL	LC-MS/MS (ESI(+))	[31]
			A: 0.15 ng/mL		
			D: 0.10 ng/mL		
Serum	NA, A, D	Fluorescent graphene quantum dots	NA: 5 µM	Transmission electron microscopy (TEM)	[32]
			A: 0.7 µM		
			D: 0.007 µM		

**Table 3 molecules-28-02900-t003:** Accuracy and precision results established on synthetic urine samples.

Analyte	QC, ng/mL	Inter-Day (*n* = 6)		Intra-day (*n* = 6)		Linear Range, ng/mL	R2
		Accuracy, %	Precision, %	Accuracy, %	Precision, %		
PG-octopamine	250	1.7	3.7	1.3	4.1	5–500	0.993
	100	3.6	5.4	3.2	6.8		
	10	8.2	13.3	7.4	12.2		
PG-dopamine	250	1.9	4.2	1.5	4.9	5–500	0.991
	100	4.1	6.3	3.6	7.5		
	10	8.8	14.1	8.1	14.7		
PG-adrenaline	250	2.2	4.0	1.5	4.6	5–500	0.991
	100	4.7	6.1	3.5	7.3		
	10	8.3	13.8	7.8	14.5		

**Table 4 molecules-28-02900-t004:** Comparison of sensitivities obtained with different derivatization agents.

Analyte	Derivatization Reagent	t_R_, min	LOD, ng/mL	LOQ, ng/mL	Theoretical Mass, *m*/*z*	Observed Mass, *m*/*z*	Mass Error, ppm
Adrenaline	PG-Cl	3.25	1.5	5	393.1057 (+Na^+^)	393.1059 (+Na^+^)	−0.51
	DNS-Cl	n.d.					
	FMOC-Cl	4.92	2.5	5	428.1468 (+Na^+^)	428.1455 (+Na^+^)	3.04
Octopamine	PG-Cl	3.21	1.5	5	363.0951 (+Na^+^)	363.0943 (+Na^+^)	2.20
	DNS-Cl	3.83	3	10	387.1373 (+H^+^)	387.1366 (+H^+^)	1.81
	FMOC-Cl	4.81	5	10	398.1363 (+Na^+^)	398.1353 (+Na^+^)	2.51
Dopamine	PG-Cl	3.35	1.5	5	363.0951 (+Na^+^)	363.0941 (+Na^+^)	2.75
	DNS-Cl	3.93	3	10	387.1373 (+H^+^)	387.1366 (+H^+^)	1.81
	FMOC-Cl	5.22	25	50	398.1363 (+Na^+^)	398.1352 (+Na^+^)	2.76
Gabapentin	PG-Cl	3.97	-	-	381.1421 (+Na^+^) 359.1601 (+H^+^)	381.1414 (+Na^+^) 359.1601 (+H^+^)	1.84 0.00
	DNS-Cl	n.d.					
	FMOC-Cl	5.82	-	-	416.1832 (+Na^+^)	416.1824 (+Na^+^)	1.92

n.d.—not detected.

**Table 5 molecules-28-02900-t005:** Analysis of real urine samples using FMOC-Cl and PG-Cl as derivatization reagents.

Sample Number	Derivatization Reagent					
	FMOC-Cl			PG-Cl		
	Adrenaline, ng/mL	Dopamine, ng/mL	Octopamine, ng/mL	Adrenaline, ng/mL	Dopamine, ng/mL	Octopamine, ng/mL
Sample 1	51 ± 8	223 ± 29	58 ± 9	57 ± 9	214 ± 27	63 ± 10
Sample 2	62 ± 10	248 ± 34	64 ± 10	65 ± 10	255 ± 35	71 ± 12
Sample 3	47 ± 7	212 ± 27	52 ± 8	54 ± 8	226 ± 29	58 ± 9

## Data Availability

Data available within the article.

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
