# Peer review of "Phthalylglycyl Chloride as a Derivatization Agent for UHPLC-MS/MS Determination of Adrenaline, Dopamine and Octopamine in Urine"

_molecules, 2023, doi:10.3390/molecules28072900_

Round 1

Reviewer 1 Report

Lines 113-114: what is the maximum efficiency of derivatization? How did they calculate this efficiency, what are the criteria for determining that it is maximum?

There is no point in presenting the gradient in the table, please put it in the description of the experimental part. Table 3 should be removed.

Gradient elution: do not understand the gradient program. Compounds are eluted before 4 minutes, and after 4 minutes, an organic solvent increases to 90%. Why is that? Besides, why is there 5% solvent between 0-1 minute, and then we have a build-up to 60% in 1.7 minutes? Analogous results will be obtained if the authors make a linear gradient from 0 to 60% in 2.7 minutes.

The authors repeatedly emphasize that these analytes have no retention in RP HPLC, and their derivatization increases retention and separation. In the scientific literature, many examples contradict the authors' claim. These compounds can be successfully analyzed by RP HPLC without derivatization. Moreover, separation and qualitative and quantitative analysis is possible. Derivatization is not necessary (check https://doi.org/10.3390/separations9080224; 10.1002/bmc.4978; 10.1081/JLC-120025598; https://doi.org/10.1155/2021/8821126). Maybe it is enough to write in the publication that the purpose of the research was to test a new derivatizing reagent.

Figure 6, lines 150-153: the authors write that the derivatization greatly increased retention, to be able to really evaluate this, a chromatogram should be placed for the separation of these compounds before derivatization.

Author Response

Notice 1:

Lines 113-114: what is the maximum efficiency of derivatization? How did they calculate this efficiency, what are the criteria for determining that it is maximum?

Response 1:

Dear Sir, thank you for your notice! As you can see in fig. 3 and fig. 4, the first step was an optimization of the derivatization temperature value. In fig. 3, it can be seen that 30 °C is enough to initialize the reaction and achieve a high yield of the derivatives. However, a 2-hour thermostating was used to exclude the possibility of the partial efficiency, which is undesirable for an analytical procedure. In routine analysis we need quick and efficient reactions, that is why the second step was investigation of the sufficient reaction time. As you can see at fig. 4, the optimal value was 10 min. Almost the same peak area was observed after 2 hours for all analytes, so there is no need to wait for 2 hours.

 Notice 2:

There is no point in presenting the gradient in the table, please put it in the description of the experimental part. Table 3 should be removed

Response 2:

Agreed. The table was eliminated, description was added in experimental section (4.2). We also added the missed information about sample injection volume (10 µL).

Notice 3:

Gradient elution: do not understand the gradient program. Compounds are eluted before 4 minutes, and after 4 minutes, an organic solvent increases to 90%. Why is that? Besides, why is there 5% solvent between 0-1 minute, and then we have a build-up to 60% in 1.7 minutes? Analogous results will be obtained if the authors make a linear gradient from 0 to 60% in 2.7 minutes

Response 3:

Dear Sir, thank you for your notice! Yes, you are right that the same results could be achieved using short gradient, but in the case of its application to urine samples, we cannot stop the elution program at 60%, because such endogenous compounds as steroid hormones require strong eluent (approx. 80-90% of methanol) with further column regeneration and equilibration before next injection. The first step with 5% is required to prevent phase de-wetting. It should be also noticed that 1 min at a weak eluent was used to exclude the entrance of dissolved salts from urine into the mass spectrometer (the 6-port diverter valve to the mass spectrometer was in the waste position during the first minute) and, at the same time, we calibrated Q-TOF using ammonium formate injection via the 6-port diverter valve.

Notice 4:

The authors repeatedly emphasize that these analytes have no retention in RP HPLC, and their derivatization increases retention and separation. In the scientific literature, many examples contradict the authors' claim. These compounds can be successfully analyzed by RP HPLC without derivatization. Moreover, separation and qualitative and quantitative analysis is possible. Derivatization is not necessary (check https://doi.org/10.3390/separations9080224; 10.1002/bmc.4978; 10.1081/JLC-120025598; https://doi.org/10.1155/2021/8821126). Maybe it is enough to write in the publication that the purpose of the research was to test a new derivatizing reagent.

Response 4:

Dear Sir, thank you for your notice! Yes, our main mistake in introduction – we did not mention that the analytes poorly retained on conventional C18 sorbents, like ours. In this case, analytes were eluted in dead volume.

We also added information about possibility of mixed-mode C18 columns usage and columns with indexes “Polar” and “Aqua”, because they can be used with 100% aqua eluent without phase de-wetting.

This is a very important and valuable notice, we added information and included provided references! Thank you!

Notice 5:

Figure 6, lines 150-153: the authors write that the derivatization greatly increased retention, to be able to really evaluate this, a chromatogram should be placed for the separation of these compounds before derivatization.

Response 5:

Dear Sir, thank you for your notice! Unfortunately, in the case of conventional C18 sorbent they cannot be separated, and they are eluted in dead volume (which is not recorded in our conditions since the 6-port injection valve is in the waste mode). Previously, we investigated this possibility using the same column and elution conditions (ref. 33). But it is not a problem to make few runs under these conditions to show absence of the retention if it necessary.

Reviewer 2 Report

The paper desribes the LC/MS/MS analytical method with phthalylglycyl derivatization for adrenaline, dopamine, and octopamine in urine.

The derivatization process is very simple and rapid, and so it is useful for routine analysis.

The method validation data is also adequate accuracy and it has a great reproducibility.

I have a few comments and questions.

1) Conclusion; The authors state the high sensitivity of this method. However, it is not easy to know it from Table or Figures.

2) Figure 3 and 4; The authors should added the error bars on the bars.

3) Figure 6; The Y-axis title "Intensitivity" is a misspelling of "intensity".

4) Table 6; The quantitative values has no need to be shown. The values are shown as the average values.

Author Response

Notice 1:

Conclusion; The authors state the high sensitivity of this method. However, it is not easy to know it from Table or Figures.

Response 1:

Dear Sir, thank you for your notice! The comparison of the sensitivity was presented in table 5. It is about 2 times higher than in the case of FMOC-Cl and DNS-Cl. If it is necessary, we can add this information in conclusion. The only one reason of this information absence in the manuscript – we tried to make it as a short and clear investigation.

Notice 2:

Figure 3 and 4; The authors should added the error bars on the bars.

Response 2:

Thank you for your notice! Proper corrections were done, error bars were added!

Notice 3:

Figure 6; The Y-axis title "Intensitivity" is a misspelling of "intensity".

Response 3:

Thank you for your notice! Proper corrections was done!

Notice 4:

Table 6; The quantitative values has no need to be shown. The values are shown as the average values.

Response 4:

Dear Sir, in this table, we presented the results of the urine quantitative analysis to show the applicability of PG-Cl as a derivatization reagent and comparison with FMOC-Cl. The aim of this table is to show the absence of significant difference between results. Should we eliminate measurement error value in this table?

Round 2

Reviewer 1 Report

The authors have responded comprehensively to all the reviewer's comments. The reviewer agrees with the authors' arguments and believes that the publication can be published.